
# Assessing the effectiveness and the economic impact of evacuation: the case of Vulcano Island, Italy

Costanza Bonadonna[1], Ali Asgary[2], Franco Romerio[3,4], Tais Zulemyan[1], Corine
Frischknecht[1], Chiara Cristiani[5], Mauro Rosi[6], Chris E. Gregg[1,7], Sebastien Biass[8], Marco
Pistolesi[6], Scira Menoni[1,9], Antonio Ricciardi[5]

[1]Department of Earth Sciences, University of Geneva, Geneva, Switzerland
[2]School of Administrative Studies, York University, Toronto, Canada
[3]Geneva School of Economics and Management, University of Geneva, Geneva, Switzerland
[4]Institute for Environmental Sciences, Geneva, Switzerland
[5]Dipartimento della Protezione Civile, Roma, Italy
[6]Dipartimento di Scienze della Terra, Università Pisa, Italy
[7]Department of Geosciences, East Tennessee State University, Johnson City, USA
[8]Earth Observatory of Singapore, Nanyang Technological University, Singapore
[9]Politecnico di Milano, Architettura e Pianificazione, Milano, Italy

*Correspondence to*: Costanza Bonadonna (costanza.bonadonna@unige.ch)

**Abstract**

Evacuation planning and management represents a key aspect of volcanic crises because it can increase people protection as well as minimize the potential impact on the economy, properties, and infrastructure of the affected area. Assessment of evacuation scenarios that consider human and economic impact is best done in a pre-disaster context as it helps authorities develop evacuation plans and make informed decisions outside the highly stressful time period that characterizes crises. We present an agent-based simulation tool that assesses the effectiveness of different evacuation scenarios using Vulcano island (Italy) as a case study. Simulation results show that the overall time needed to evacuate people should be analysed together with the percentage of people evacuated as a function of time and that a simultaneous evacuation on Vulcano is more efficient than a staged evacuation. We also present a model to assess the economic impact of evacuation as a function of evacuation duration and starting period that reveals that an evacuation of Vulcano would cause significant economic impact to the tourism industry if lasting more than 3 months (in case it was initiated at the beginning of the visitor season) to 1 year (in case it was initiated at the end of the visitor season).

**Keywords**: volcanic crisis, emergency evacuation, evacuation modelling, staged evacuation, simultaneous evacuation, evacuation effectiveness, La Fossa volcano

## 1. Introduction

Evacuation is a key measure used in emergencies that can save lives and reduce human impact (e.g. Moriarty et al., 2007; Tomsen et al., 2014; Thompson et al., 2017). The call for evacuation is often taken



under pressure and uncertainty (Bebbington and Zitikis, 2016) and is a costly decision which, depending on how it is managed, can lead to both positive and negative outcomes (Doyle et al., 2014). Miscalculation or delays in the key phases of the evacuation process such as the timing when the evacuation order is issued, the channels and sources through which the order is communicated to the public, the time required by the population to process the evacuation order and their actions, and

evacuation logistics and routes, can significantly reduce the evacuation effectiveness (Sparks, 2003; Sorensen and Sorensen, 2007; Lindell et al., 2019). Evacuation planning carried out in a pre-disaster context provides a better understanding of when to issue evacuation orders, who should be evacuated at what time, which routes and alternate routes should be considered, where evacuees should go, what resources are needed and how long the evacuation might last (MCDEM 2008; Marzocchi and Woo,

2009). In this regard, volcanic crises differ from other natural hazards as they are often associated with an *unrest* phase, during which most volcanic systems exhibit precursors from hours to days, weeks, and even months before the onset of an eruption (Gregg et al., 2015). These unrest phases are associated with long-lasting, time-dependent uncertainties regarding forecasts on where, when, and even if a future eruption will take place. However, volcanic unrest represents an important phase during which

preparedness activities can, and should, be initiated with the aim of increasing resilience of the system as well as facilitating the potential evacuation.

Evacuation orders can be issued before the actual eruption onset in case increasing volcanic unrest is observed. This type of evacuation is an important preventive measure and an efficient response to minimize human impacts (Wilson et al., 2012; Baxter et al., 1998; Leone et al., 2019). If planned and

implemented well, preventive evacuation can save lives as was the case in many past volcanic emergencies, including the 1991 eruption of Pinatubo volcano in the Philippines, the 2006 and 2010 Merapi eruption in Indonesia, the 2017-2019 eruption of Mt. Agung in Bali and the 2017-2018 eruption of Manaro Voui in Vanuatu. Evacuation can also be initiated after the beginning of an eruption, especially in case of short or no warning. In these situations, people may have to be evacuated due to

the approaching of potentially impactful hazards such as lava flows, pyroclastic density currents (PDCs), lahars and tephra fall. However, failure to evacuate in anticipation of an eruption or of the associated primary and secondary hazards can lead to catastrophic outcomes as seen during the 1985 Nevado Del Ruiz eruption in Colombia and during the 2018 Fuego eruption in Guatemala (Voight et al., 2013; Leone et al., 2019).

Unlike other emergencies, the duration of volcano-related evacuations can last for days, months or even years depending on the type of eruption and its impacts on the landscape and can result in long-lasting or even permanent relocation of communities (e.g., Soufrière Hills, Montserrat and Tungurahua, Ecuador; Barclay et al., 2019). In detail, long duration of evacuation occurs mostly because i) elevated unrest can be protracted, ii) eruptive activity can be protracted, iii) post-eruption activity such as

remobilisation of pyroclastic deposits by water (i.e. lahars) and wind (i.e. ash storms) can continue




threatening communities, and/or iv) the damage can be so overwhelming that people and their government lack the resources to rebuild in a timely period.

While evacuations can save lives, they are costly and may trigger other adverse economic and social impacts (e.g. Bouwer et al., 2011), especially in the context of false alarms (e.g. Woo, 2008). 80 Additionally, the consequences of a certain hazardous phenomenon can be lower than predicted. As an example, since the 1950s, 75% of evacuations issued due to tsunami warnings turned out to be either false alarms or the generated tsunami was not as impactful as expected (The Economist 2003; Selva et al. 2021). In this context, the potential economic impacts of an eruption should be accounted for in the process of decision-making for evacuations. According to Meyer et al. (2013), the management of 85 natural risks can result in five different types of costs (**Table 1**). First, direct costs that result from the

| | | Tangible Costs | Intangible (non-market) Costs |
|---|---|---|---|
| **Damage Costs** | Direct | Physical damage to assets: e.g. buildings, infrastructure | Loss of life Health effects Loss of environmental goods |
| | Business Interruption | Production interruption because of destroyed machinery | Ecosystem services interrupted |
| | Indirect | Induced production losses of suppliers and customers of companies directly affected by the hazard | Inconvenience of post-hazard recovery Increased vulnerability of survivors |
| **Risk Mitigation Costs** | Direct | Set-up of mitigation infrastructures Operation and maintenance costs of those infrastructures | Environmental damage due to the development of mitigation infrastructure or due to a change in agricultural practices |
| | Indirect | Induced costs in other sectors due to the disruption caused by the mitigation measures | Impact on the well-being due to the disruption of services |

**Table 1 Cost categorizations with examples (modified from Meyer et al, 2013)**

physical destruction of assets due to the interaction with hazards. Second, business interruption costs that refer to losses that occur in areas directly affected by the hazard when people are not able to carry out their work because the workplace is destroyed, damaged or not accessible. Third, indirect costs that are induced by either direct damages or business interruption costs (e.g. production losses for suppliers 90 and customers of entreprises); they can occur inside or outside of the hazard zone. Fourth, intangible costs that concern damages to goods and services for which market prices do not exist such as the impacts on environment, health or cultural heritage. Fifth, risk mitigation costs, which include risk management planning and adaptation plans, hazard modification, monitoring and early warning, emergency response and evacuation. This category itself can also be divided in subgroups such as direct 95 (any action taken for mitigation infrastructures), indirect (secondary costs such as economic disruption due to mitigation measures) and intangible (e.g. environmental damage due to change in agriculture practices) costs (Meyer et al., 2013). In this study, we are mostly concerned with the third type of cost related to the interruption of economic activities (i.e. tourism) as a result of a prolonged evacuation.





In this context, we present here a novel methodology to couple an evacuation model with an assessment of its potential economic impact. We use the island of Vulcano, Italy, to illustrate strategies for the assessment of the effectiveness of an evacuation as well as its economic impact on the main source of revenue, i.e. tourism. In the past decade, evacuation and civil protection planning have been underway in Italy for the main active volcanoes (e.g. Vesuvius and Campi Flegrei, www.protezionecivile.gov.it; Baxter et al., 2008; Marzocchi and Woo, 2009). However, with the exception of Stromboli Island after the volcanic-induced tsunami in 2002, limited planning efforts have been carried out for the other volcanic systems in the Aeolian Islands, an archipelago in the South of Italy composed of seven islands. The Aeolian islands, which earned UNESCO World Heritage status in 2000, are a volcanic arc associated with the subduction of the African plate under the Eurasian plate, of which Stromboli volcano on the island of Stromboli and La Fossa volcano on the island of Vulcano are the youngest and most active volcanic edifices (Selva et al., 2020).

We developed an agent-based simulation in GIS space using the AnyLogic® software platform to assist emergency managers and assess the effectiveness of specific evacuation parameters, i.e. number of people present on the island (during the low and high seasons), type of evacuation (simultaneous whole community evacuation or sequentially staged evacuation of different areas), eruption probability, exposure, timing (before, during or after the eruptive event). A strategy to assess the economic impact of an evacuation based on the analysis of the consequences on the main economic activity (i.e., tourism) is also presented.

The next section provides some conceptual background related to effective evacuation, types of evacuation methods, and evacuation modelling, while section three describes the study area. Section four illustrates the methodology adopted in our analysis, while sections five and six present and discuss the results on the assessment of evacuation efficiency as well as the assessment of economic impact of an evacuation considering different durations of evacuation and evacuation of different areas. Section seven provides conclusions.

## 2. Background on effective evacuation

Han et al. (2007) developed and described a four-tier evacuation effectiveness framework, by looking at evacuation time, individual evacuation time, exposure over time, and spatio-temporal exposure measures. Effectiveness of evacuation planning and operations for volcano emergencies can be assessed using this four-tier framework. One of the most common goals of evacuation analysis and planning is to improve the effectiveness of evacuation by reducing evacuation time to minimize the adverse impacts associated with people leaving their place of employment, study, or their homes. Several methods have been proposed to improve the effectiveness of emergency evacuation such as enhancing evacuation order and warning dissemination, controlling flows and movements in and out of designated areas, implementing staged evacuation, directing people to the best evacuation routes, and focusing on





flexibility to plan a possible evacuation (Abdelgawad and Abdulhai, 2009; Gaudard and Romerio, 2015).

In this paper we distinguish between "evacuation time", defined as the time required for the last person to evacuate an emergency zone (Urbanik, 2000), and "evacuation duration", which represents the period during which a community has been removed from a risky area. In addition, we define "evacuation
effectiveness" as the time required to evacuate a certain fraction of the population (e.g., 95%) (Han et al., 2007). Evacuation time of individuals or families depends on a number behavioural, logistical, perceptual, and communication factors (Tomsen et al., 2014). In order to minimize evacuation time, it is, therefore, important to reduce evacuation warning time (time it takes for the evacuation warning to reach each individual), evacuation preparedness time (time it takes for individuals to prepare for
evacuation after receiving evacuation warning), and evacuation travel time (time it takes for individuals to travel from residence to evacuation designated areas). Each of these time segments varies from person/family to person/family depending on their demographic attributes, preparedness levels, and access to information and resources (Jumadi et al., 2019, Lechner and Rouleau, 2019).

**2.1 Simultaneous and Staged Evacuations**

Evacuation can be implemented using simultaneous or staged methods. In the simultaneous evacuation, people in an exposed area are informed and expected to evacuate simultaneously. In the staged evacuation, the exposed area is divided into several zones, and people in each zone are evacuated according to a specific order (Sbayti and Mahmassani, 2006). Both simultaneous and staged evacuations
have been used in past emergencies. Staged evacuations have been frequently used during hurricanes and for the 2002 Los Alamos wildfire in New Mexico (Malone et al., 2001; Farrell, 2005; Wolshon et al., 2006). Simultaneous evacuations are often used during sudden emergencies when rapid evacuation is necessary (e.g., earthquake, landslide, industrial accidents), whereas staged evacuation is more effective when sufficient lead time exists to prepare for evacuation or when resources are limited for
simultaneous evacuation of the whole population. Chen and Zhan (2008) found that simultaneous evacuations are more suited in areas of low traffic congestion, whereas staged evacuation may be the most effective in high population density areas and complex street networks. In case of staged evacuation, the number of stages can influence the evacuation effectiveness and thus optimising the number of stages is essential in reducing delays during the evacuation process (Chien and
Korikanthimath, 2007). Jumadi et al. (2019) developed a staged evacuation using a spatial multi-criteria analysis for prioritisation of evacuees and found that while the staged evacuation was more effective in reducing potential traffic congestion, the simultaneous evacuation still showed better results in reducing the population at risk.


## 2.2 Evacuation Modelling and Simulation

Several simulation and modelling approaches have been proposed and used for evacuation including cellular automata, game theoretic, discrete events, multi-criteria decision support systems (Cole et al., 2005; Marrero et al., 2013), agent-based (Voight et al., 2000; Carver and Quincey, 2005; Jumadi et al., 2016), and experimental methods (Yang et al., 2015). Evacuation modelling has been performed for small and medium scale emergencies such as building fire, structural blast (Pluchino et al., 2015), metro stations (Wang et al., 2013), oil and gas platforms and factories (Cheng et al., 2018), university campuses (Asgary and Yang, 2016). Larger scale emergency evacuations have also been modelled, such as volcanic eruption, flooding, and hurricane (Jumadi et al., 2016; Bernardini et al., 2017; Fahad et al., 2019).

Agent-based modelling (ABM) is emerging as a suitable and promising framework for evacuation analysis and planning in recent years (Chen and Zhan, 2008; Liang et al., 2015; Jumadi et al., 2019). ABM is appropriate for modelling complex and interactive systems (Gilbert and Bankes, 2002) such as emergency evacuation because it combines behavioural attributes with spatial and environmental data (Brown and Xie, 2006). Moreover, ABM can provide a more realistic evacuation simulation with respect to aforementioned approaches by incorporating human agents to the geographical environment (Mas et al., 2012; Joo et al., 2013).

## 3. Case Study: Vulcano island, Italy

Vulcano is the southernmost of the seven Aeolian Islands located in the Tyrrhenian Sea (25 km north of Sicily). It has a surface area of ~20 km$^2$ and contains five main settlements, i.e. Vulcanello, Porto, Lentia, Piano and Gelso (**Fig. 1**). The areas of Vulcanello and Porto both have mixed land use zones with commercial, residential, and tourism activities. Piano is mostly a residential area with one dual elementary/middle school (for children up to 14 years in age). Lentia and Gelso are small residential areas (associated with <4% of total residents on Vulcano). Vulcano has a few local critical facilities and infrastructures that include three helipads, one main port (Porto Levante) and two smaller ports (Porto Ponente and Gelso), one main power plant in Porto and one secondary solar plant in Piano as well as one telecommunication station, one desalination plant and one waste-water plant in Porto. The road network is limited with only one road connecting Porto and Piano (Galderisi et al., 2013; Bonadonna et al., 2021).

Vulcano's predominant economic activity is tourism (Galderisi et al., 2013; Aretano et al., 2013; Bonadonna et al., 2021). The island's economy and urbanization have been growing fast since the 1980s by attracting tourists from Italy and other countries, particularly during the summer season. Vulcano has a floating population passing from about 800 residents in the winter to monthly peaks of about 22,000-28,000 visitors in July-August (Bonadonna et al., 2021). With increasing number of visitors and seasonal workers, volcanic risk also increases and, therefore, emergency management, particularly evacuation planning and preparedness, has become an important issue for the island.



**Figure 1. a) Built up areas, critical infrastructure and b) economic activities on Vulcano Island.
In the inset the location of Vulcano island in relation to mainland Sicily and the closest large port
Milazzo is also shown.**



### 3.1. Geological settings and implications for evacuation plannning

In terms of geological settings, Vulcano consists of several overlapping volcanic structures including two caldera systems (i.e. Il Piano caldera to the south and La Fossa caldera in the central portion of the island), and a smaller structure (i.e. Vulcanello) in the northern side of the island. A stratovolcano (i.e., La Fossa cone) sits within the La Fossa caldera and three smaller and coalescing pyroclastic cones sit atop the Vulcanello islet. Subaerial volcanic activity in the island dates to 135 and 120 ka (Zanella, et al., 2001), with La Fossa cone (hereafter referred to simply as La Fossa) starting at ~6 ka and being the current most active system (Dellino et al., 2011). The last eruption of La Fossa was a long-lasting Vulcanian cycle that occurred between 1888-1890 (Mercalli and Silvestri, 1891). The eruption produced emission of ballistics, tephra fallout, and intense remobilization of tephra-fallout deposits by rainwater into lahars (Di Traglia et al., 2013). However, it is also important to consider that the activity of La Fossa has been characterized by a large variety of eruption styles, including effusive activity and explosive events. Among this variety, hydrothermal events of various intensity have occurred and associated with impactful hazards such as blast, diluted PDCs and ballistic fallout, with the most violent being the Breccia di Commenda eruption dated around 1230 (Rosi et al., 2018). It is thus important to distinguish between magmatic events, for which the main driver is the magma rising to the surface, and hydrothermal (or phreatic) events, for which the main driver is the interaction amongst water, rocks, and magmatic heat and gas (e.g. Barberi et al., 1992; Rouwet et al., 2014; Stix and de Moor, 2018). By their nature, hydrothermal events may be more difficult to predict than magmatic unrests and they can also happen outside the main active vent (as it has often been the case at La Fossa). In fact, La Fossa system is a permanent and powerful emitter of fluids whose flow is maintained by an elevated gas over-pressure in the subsoil (Selva et al., 2020). Even modest imbalances in the supply of fluids can trigger explosive eruptions as the numerous cases that have occurred on Vulcano after the Breccia di Commenda in the last eight centuries demonstrate (e.g. 1444 AD, 1550 AD, 1727 AD, 1873-76 AD; Selva et al., 2020). This is also the case of other volcanoes that have been associated with recent and sudden explosions such as White Island and Tongariro in New Zealand (Breard et al., 2015; You Lim and Flaherty, 2020), Ontake in Japan (Oikawa et al., 2016) and Turrialba and Poas in Costa Rica (Alvarado et al., 2016; de Moor et al., 2016). While at most volcanoes mentioned above that are located in remote areas these hydrothermal events represent a threat mostly for tourists, in Vulcano they represent a serious threat also for inhabitants that live very close to the volcano (e.g. Porto area on the north of La Fossa). The main infrastructures (including two ports - Porto di Levante and Porto di Ponente, the telecommunication station, and the main power plant) and the majority of economic and touristic activities are concentrated in the Porto area also located just north of La Fossa cone (**Fig. 1**). This is why in the case of Vulcano the potential for evacuation becomes an important issue even in case of weak unrest. Both a hydrothermal explosion and a magmatic eruption would be especially challenging if they happened during the high season (July-August) and with little or no warning, as it has been the case for the recent small but deadly eruptions at touristic places mentioned above.


Although it would appear to be a quick and small operation, evacuation of the island under different
        weather and marine conditions, occurrence of different hazards (e.g. tephra fall, PDCs, lava flows,
        lahars, landslides, tsunami) and various seasons (summer versus winter) could result in different
        decisions and actions. Moreover, one must also account for unforeseen factors that might limit the
        availability and efficiency of evacuation (e.g. damaged harbours). Scientists including those of the

Istituto Nazionale di Geofisica e Vulcanologia (INGV) and of the Consiglio Nazionale delle Ricercheof
        Italy - Institute for electromagnetic sensing of the environment (CNR-IREA) continuously monitor all
        active volcanoes in Italy, including La Fossa, periodically transferring information on the state of
        volcanic activity to the national and regional decision makers each with defined authorities, roles and
        responsibilities that are part of the overall Civil Protection system in Italy (Legislative decree "Codice

della protezione civile", 2018). Evacuees may include local residents, national and international tourists,
        and seasonal workers. While a general municipal plan for emergency management on Vulcano exists
        (http://www.comunelipari.gov.it/zf/index.php/servizi-aggiuntivi/index/index/idservizio/20015), and an
        evacuation drill was carried out with the residential population in 1991 after a period of seismic unrest
        at La Fossa, a detailed and updated evacuation plan for the Vulcano island does not currently exist.

Recently, the Italian Civil Protection Department has undertaken a dedicated effort to finance detailed
        studies  of the current understanding of the volcanic system and of the whole range of potential volcanic
        hazards (Selva et al., 2020). Based on these results the alert level system is being reviewed in
        collaboration with the scientific community.

**4. Methods**

        **4.1. Agent-Based Modelling of pedestrian evacuation**

        The Vulcano evacuation simulation tool has been developed using the Anylogic platform (version
        8.7.5), which provides ABM capabilities as well as GIS spatial data incorporation. Our simulation tool
        includes four main agents, each of which is described below (i.e. Hazard, Evacuees, Ferries and Ports,

Agents' Environment). In order to correctly characterize such agents and tailor the analysis to the
        specifics of the island without which the tools would be useless, risk factors including hazard,
        vulnerability and exposure of both the community and critical assets must be known as well as their
        dynamics over time during the year and the different seasons. Such key elements have been extensively
        addressed and analyzed in a paper presenting a novel risk assessment model for volcanic risk, named

ADVISE, based on long term research efforts of the authors and applied to the Vulcano Island
        (Bonadonna et al., 2021). All the needed aspects and elements required to assess the various indicators
        are provided there and rely on an extended work of surveys and data collection carried out in the last ten
        years. The following ABM uses the outcomes of such data collection and risk assessment, especially as
        far as hazard and exposure are considered. Some aspects of systemic vulnerability are also considered

related to the accessibility of the three ports of the Island and their intrinsic characteristics.





### 4.1.1 Hazard (volcano) agent

We define La Fossa volcano as a physical agent with specific behaviour and states. In this ABM, La Fossa volcano has three main states including background level (i.e. normal conditions), unrest (devided

into Attention, Pre-Alarm, Alarm) and eruption (**Fig. 2**). It is important to consider that, for simplicity, the states in this ABM are general states and do not correspond to the alert system specific to Vulcano, which is still under evaluation. However, modifications can be made to reflect the specificities of individual volcanoes. Normal conditions vary for each volcano; in Vulcano, they consist of fumarolic emissions (mostly concentrated in two main fumarolic fields located in the northern rim of the active

crater of La Fossa cone and at the beach of Baia di Levante), ground deformation, earthquakes and accompanying landslides (Barberi et al., 1991; Selva et al., 2020). While the volcano agent can be very complex, here we only include the volcano behaviours and states that impact the evacuation process. As such we assume that the simulation starts when the volcano is in the Alarm state (**Fig. 2**). In the Alarm state, eruption is assumed to be imminent or highly likely such that a mandatory evacuation order is

issued.

Shift from Alarm state to the Eruption state is handled through a condition transition that is linked to a user-defined table function. Forecasting of a volcanic eruption can be based on functions discussed in literature, e.g. exponential hazard function (Ho, 1992; Cornelius and Voight, 1994; Chastin and Main, 2003; Connor et al., 2003; Cruz-Reyna and Reyes-Dávila, 2001). The spatial exposure of evacuees can also be determined based on the probability of being impacted by an eruption using Aucker et al. (2013) and Brown et al. (2017) models as well as hazard analyses of Vulcano (Dellino et al., 2011; Biass et al., 2016a, 2016b; Gattuso et al., 2021). The probability of being impacted by various volcanic hazards depends on eruption

**Figure 2. State chart of the volcano agent**

dynamics (i.e., occurrence of ballistics, tephra fallout, lava flows, blast surge-like PDCs, lahars) as well

as topography and atmospheric conditions (e.g., wind speed and direction). However, given that before the actual eruption (hydrothermal or magmatic) takes place, the extent and intensity of the associated hazards are not known, we consider here the evacuation of certain areas to be based on the worst-case scenario, e.g. occurrence of PDCs and ballistic ejection in the case of Vulcano, which could impact the





whole La Fossa Caldera (including Porto area) and part of Piano Caldera (e.g. Dellino et al., 2011; Biass
et al., 2016b).

### 4.1.2. Evacuee Agent

We combined and expanded the Sorensen and Mileti (2014) and Stepanov and Smith (2009) multi-step
evacuation process models to include four main time segments: 1) warning issuance, the step from when
unrest or evidences of hazard appear to when decision makers decide to issue the warning; 2) warning
diffusion, the process from when the warning is issued to when the warning reached the intended
audiences; 3) evacuation decision and preparation; and 4) evacuation movement. A statechart is used to
model the evacuee agent's evacuation behaviour (**Fig. 3**). The agent is created, and its initial state is set
to "before warning" (or normal). As soon as a warning is issued, the agent's states change from "before
warning" to "warning issued", corresponding to an evacuation order. Transition from this state to the
"warning received" is controlled by a time out triggered transition. We use a normal truncated

**Figure 3. State chart of evacuee agent**

distribution for this transition
with minimum and maximum
time values that can be set by
evacuation planners before
running the simulation. Use of
this distribution allows us to limit
the lower bound to 0 and the
upper bound to a finite value.
Transition from the warning
received state to prepared state is
also handled by a truncated
normal distribution that can be set
by the evacuation planner.
However, this transition is
triggered only if the evacuation is
either a simultaneous evacuation
or the evacuee is located in the

assigned evacuation stage. The order of evacuation during a staged evacuation is based on the proximity
to the hazard, with the most exposed people being evacuated first. In Vulcano, the North part of the
Island will be evacuated from the Levante and Ponente ports and the South of the island from the port
of Gelso. In our simulations, people in Porto and Piano will be simultaneously evacuated first, and
people in Vulcanello will be evacuated last. However, to provide the emergency planner with more
flexibility, the simulation allows the users to set the evacuation order as needed. Evacuation time
depends on the evacuees' pedestrian speed and their distance to the closest active port. We consider the


walking speed as a uniform distribution, but the model allows the lower and upper bounds of this distribution to be set depending on the environmental situations and population scenarios being analyzed. We assume here only pedestrian evacuation, but the simulation can be adapted to also include evacuation by vehicles, or a combination of the two. We recognize that while walking may be a more

feasible option for those in the north part of the Island, it may be more difficult for the people in the south part of the island. Upon arriving at the closest active port, evacuees wait for ferries. Once the ferries arrive, evacuees board and they are considered to be evacuated.

### 4.1.3 Ferries and Ports Agents

Ferries transport evacuees from ports on Vulcano southward some 44 km to the large port of Milazzo on the north shore of Sicily (**Fig. 1**). As an evacuation order is issued, available ferries are mobilized in

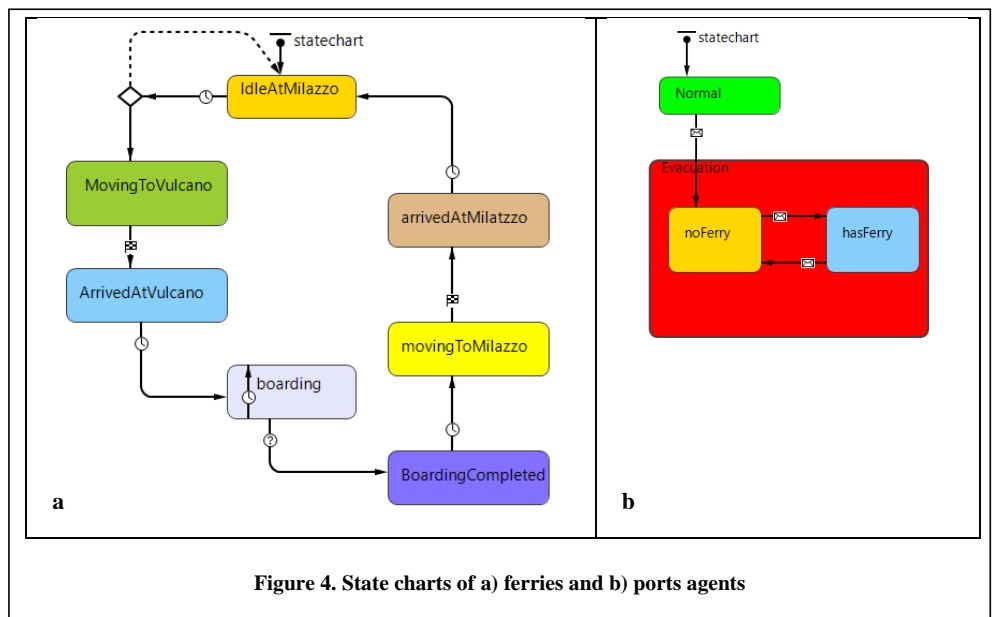

**Figure 4. State charts of a) ferries and b) ports agents**

the Milazzo port. It takes between 40 minutes to 1 hour for ferries (hydrofoils) to reach the Porto Levante in Vulcano from Milazzo. In our simulations, ferries will have a capacity ranging from 200 (hydrofoils) to 800 (ferries) passengers (including intermediate capacities of 400 and 600), but these variables can

be changed. Since the two smaller ports in Vulcano (Porto Ponente and Porto Gelso) are not suitable for large ferries, only boats with small capacities are dispatched to these ports. Boat speed depends on the weather and marine conditions that can be set by the users before running the simulation. However, for this study we use an average speed of 50 km/hour that is the regular speed of hydrofoils boats operating between Vulcano and Milazzo. As ships arrive in their port, evacuees start boarding until full capacity

is reached, at which point ships will travel back to Milazzo. If there are more requests, ferries and boats continue going back to the assigned Vulcano ports, otherwise they stay in Milazzo.


Port agents have two main states in our ABM including Normal and Evacuation states (**Fig. 4**). As soon as an evacuation order is issued, the state of the ports changes from Normal to Evacuation through a message transition. Inside the evacuation state, two substates demonstrate whether a port has ferries to

board evacuees or not. The transition between these two substates is controlled by the interactions between the ferries' agents and ports' agents.

### 4.1.4 Agents' Environment

Two main GIS networks were created for this study. The first connects the ports in Vulcano (Porto

Ponente, Porto Levante and Porto Gelso) to the port in Milazzo (**Fig. 1a**). The second connects buildings in Vulcano (residential, commercial, hotels, facilities, etc.) with the road network created based on the existing road network on the OpenStreetMap (**Fig. 1a,b**).

### 4.1.5 Model setup

We illustrate our evacuation simulation tool by setting up two pre-eruption evacuation scenarios taking place during the low and the high touristic seasons. Summary of the scenario's initial conditions are summarised in **Table 2**. **Figure 5** and **Appendix A** show the parameters and scenarios setting and the

| Parameter | Description |
|---|---|
| Warning received time (min) | Uniform distribution, between 15–90 |
| Preparedness time (min) | Uniform distribution, between 30– 120 |
| Walking speed (m/s) | Uniform distribution, between 0.8–1.6 |
| Population (low season) | Total: 1,000<br>Piano: 300<br>Vulcanello: 100<br>Porto: 600 |
| Population (high season) | Total: 4,600<br>Piano: 400<br>Vulcanello: 400<br>Porto: 3,800 |
| Number of available ferries | 10 |
| Capacity range of boats (number of people) | Hydrofoil: 200<br>Ferry: 800 |
| Average speed of ferries (km/h) | 50 |
| Evacuation order for staged evacuation | First stage: Porto (via Gelso) and Piano (via Levante and Ponente)<br><br>Second stage: Vulcanello (via Levante and Ponente) |

**Table 2: Input conditions to the model and selected values used for the scenarios considered for Vulcano. All these parameters can be adapted to the user's need.**

main interface of the Vulcano Evacuation Simulation Tool. The low season scenario involves 1,000 people consisting of 300 local residents living in Piano, 100 residents living in Vulcanello and 600

people classified collectively from residents, seasonal workers and tourists in Porto. The high season evacuation involves a total population of 4,600 consisting of 400 people each in Piano and in Vulcanello



and 3,800 people in Porto, where residents, tourists and seasonal workers are mixed across the different areas (for the sake of simplicity here we only consider Porto, Piano and Vulcanello areas as Lentia and Gelso are associated with <4% of the residents). Both scenarios assume only pedestrian evacuation, where each evacuee is assigned a walking speed uniformly sampled between 0.8 m/s to 1.6 m/s (Wood et al., 2018). The evacuation warning time follows a uniform distribution between 15 and 90 minutes and the evacuation preparedness time also has a uniform distribution ranging between 30 and 120 minutes (Table 2). In addition, our simulations do not account for variable weather and marine conditions. Note that these parameters were chosen based on the author's knowledge of the area and are used only with the purpose of illustrating the functionality of the tool. All parameters can and should be identified by emergency managers based on the availability of information and on the range of conditions to be tested (e.g. people with reduced mobility or with health issues, evacuation using a variety of vehicles).

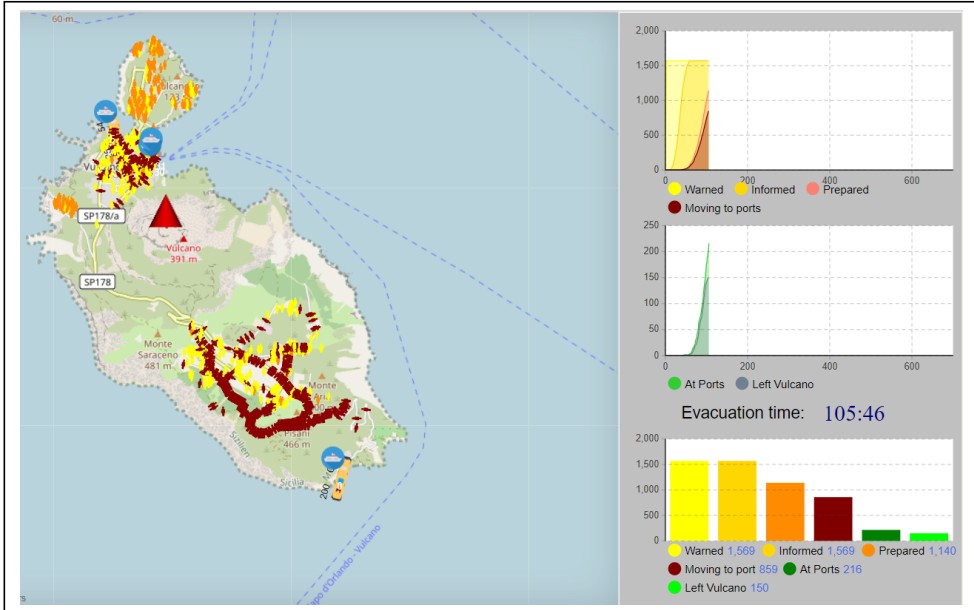

**Figure 5 Example of a pedestrian evacuation simulation run. The map on the left shows movement of evacuees from different parts of the island to their designated or nearest port from where they will be evacuated by ferry (Porto Levante, Porto Ponente in the North and Porto Gelso in the South). Yellow colour represents people who have received the evacuation order, orange colour shows people who are prepared for evacuation, and brown colour shows people who are moving towards the ports. First graph on the top right shows evolution in time of number of people warned, informed, prepared and moving to ports; graph in the middle shows number of evacuees at ports and number of people evacuated from the island with time; graph on the bottom right shows the same information as above by in a bar chart format. Evacuation time is indicated in minutes (in blue).**




### 4.2 Assessment of the economic impact of an evacuation

When the hazard level is high and human life is at stake, economic losses usually play little to no role in the decision of whether to evacuate. In less extreme situations, however, authorities weigh different factors, and different evacuation plans can be considered. In fact, the management of the crisis will take different courses depending on the evolution of the unrest and the time-dependent evolution of the hazard. Accurate data necessary for a reliable cost-benefit analysis are rarely available, especially in the context of small islands where they are aggregated at the level of the Municipality. Furthermore, in case of relatively simple economic systems such as that of a small island, complex and sophisticated models can be replaced by a set of reasonable hypotheses. Consequently, we present an approach to estimate the loss of revenue caused by a total or partial evacuation of the population on the island at any one time (i.e. residents, seasonal workers and tourists) due to an imminent eruption. Such an analysis is especially important in case of scenarios of long-lasting Vulcanian cycles, such as that of the 1888-90 eruption of La Fossa volcano, that would disrupt the island's economy for a long time (many months to years).

Data collection required to estimate the impact of an evacuation on the island's main source of revenue was carried out between 2014 and 2016. This investigation focused on tourism related business activities. We interviewed owners and workers of shops, restaurants, hotels, and a tourist office in May 2014 to constrain working seasons, business hours and consumer prices. We also spoke with the tourist office in Lipari to determine the number of tourists visiting Vulcano. This was supported with online research (2014-2016) to assess hotel prices that could not be obtained through discussions with personnel onsite. Several booking websites were used in case the hotel did not have its own website.

While there are two main beaches between Porto and Vulcanello that serve as the main attraction for visitors overall, one of the most popular touristic activities on Vulcano is the mud pool. The mudpool sits on a fault lineament between La Fossa and Vulcanello and was initially developed around an exploration drilling site for geothermal exploitation drilled in the 1950s (Faraone et al., 1986; Gioncada et al., 1995). Many people visit the island only for this reason. Tourists mostly come to the island during summer also to taste the local cuisine, to take boat tours around Vulcano and/or around other Aeolian Islands. Hiking to and around the summit of La Fossa and daily visits to other islands are also popular activities. A variety of lodging and accommodation solutions are available on the island (**Fig. 1b**).

At the time of our survey in 2014 and web search in 2016, there were 17 hotels on Vulcano with only four of them open the whole year (**Fig. 1**). Those open during the middle season are not fully occupied. On the contrary, from June until September they all operate almost at full capacity. In addition to the hotels, there were 21 B&Bs, hostels and residences with two camping areas, as well as 40 apartments. All of them are open in the high season; few are open in the middle season. Most of the restaurants are closed during the low and middle seasons. From June to September, all 24 restaurants were open until after midnight and were always full of tourists. Both day trippers from Sicily and other Aeolian Islands and visitors over-nighting on the island dine in these restaurants. Vulcano has dozens of stores located



in the Porto area, mainly consisting of clothing and souvenir shops, but very few of them are open during the whole year. The rest are open mostly around Easter until the end of October. One main supermarket and two smaller grocery stores are located in the Porto di Levante area, which is the area within the broader Porto area defined by the presence of the main port on the island (Levante) and relatively dense development. Like most of the Aeolian islands, Vulcano has many notable activities for outdoor enthusiasts. Most all leisure activities (e.g. mud pool, motor car and motor bike rental, bicycle rental, SCUBA diving) on the island are in the Porto and Vulcanello areas, although a few hiking trails exist in the Piano area and more remote area of Gelso in the far south of the island. Most activities are closed during low season, but a very few are in service during the middle season.

Three seasons have been identified based on the number of tourists, which include: Low Season (November-April) with no touristic activity on the island (most of the hotels, hostels, B&B's and residences usually undergo maintenance activities); Middle Season (April-May-June and September-October) with a gradual increase in number of tourists (some of the restaurants, hotels, hostels and B&B's open; repair of residences continues during this season); High Season (July-August) with monthly peaks that approach 22,000-28,000 visitors (e.g. 18-23 times the number of residents; Bonadonna et al., 2021). Being the closest island to Sicily, Vulcano is an easy getaway for mainland day trippers; lots of them coming to the island on their private boats and dining at the restaurants. In fact, buoys are available in Levante Bay and mooring is possible at the "Marina di Vulcanello" jetty to the north of the bay or the "Baia di Levante" jetty to the bay's south inside the commercial port. All types of leisure activities and shops are functional during the High Season.

Cheese and wine are also produced on Vulcano. The cheese factory La Vecchia Fattoria is situated in the west side of Porto and just off the road to Lentia. In 2016, the owner indicated that the farm included 280 goats, 40 cows and 30 sheep and the main production takes place between March and October, i.e. middle and high seasons. From November until February they have less goat milk because the goats are pregnant and/or feeding their lambs. While they have only a few clients in mainland Italy, exports are limited mostly to the Aeolian Islands, especially with supermarkets in Lipari where the main income is derived. The main wine factory (Punta dell'Ufala) is located in Gelso and the vineyards are dispersed on 5 hectares of slightly steep hills between Piano and Gelso. According to personal communication with the owner (Ms Paola Lantieri) in 2016, the most delicate season for the grapes is between March and July, because this is when the vineyard flourishes and becomes more susceptible to pests. They sell the wine mainly in Vulcano to hotels, restaurants and the grocery stores/supermarket and export some product to the mainland Italy and the USA and Japan rather than the other Aeolian Islands.

### 4.3 Methodology to calculate the revenues from touristic business activities in Vulcano

Our analysis focused on the turnover created by tourism-related businesses, which provides the main income to the island's economy. The turnover represents the gross revenue that a business generates without considering associated expenses (e.g., food, water, energy and maintenance). The economic



impact associated with an evacuation of the island is represented by the loss of this revenue. We do not consider the revenues from the grocery stores, supermarket and shops due to lack of sufficient reliable data nor the revenues from the cheese and wine factories because they are not tourism-related businesses. The income from maritime transport is also not included, because it does not have a major impact on the local economy.

As mentioned in **Table 1**, different categories of costs are concerned when dealing with impacts from natural events such as those involving volcanic unrest and eruption that might necessitate an evacuation of people from the island. Our focus on Vulcano was identifying tangible business interruption cost related to interruption of touristic activities. The revenues, expressed by the turnover, for all the touristic activities on the island, which will become the loss in case of an evacuation, are calculated for different

seasons as part of the cost assessment. The main touristic business on the island can be divided as B&B's, hostels and residences, restaurants and bars, hotels, leisure activities and shops. Each of these are described below.

### 4.3.1 B&Bs, hostels and residences

Data were collected from the internet and field interviews for seven B&B, one hostel and six residences (out of 21), but we were unable to obtain data for seven other structures due to lack of online information. The revenue for each season is calculated by multiplying the capacity, the price, the total days and the occupancy rate. In the equations below, H and M indicate high and medium season, respectively.

$$RH = C * PH * TH * OH \qquad \text{(Eq. 1)}$$

$$RM = C * PM * TM * OM \qquad \text{(Eq. 2)}$$

RH and RM represent the total revenues for high and middle season, respectively. C represent the total capacity, i.e. maximum number of people that can be accommodated, at a given place. PH and PM are prices per night per person; TH and TM are number of total days estimated in calculations and OH and OM are occupancy rates, i.e. the proportion of available accommodation occupied. As there are no

official statistics available, simple assumptions are made for occupancy rate that are based on observations done over more than 10 years of research on the island, which are expected to be reasonable within a margin of 5-10%. During high season, a rate of 100% is estimated and for middle season, the value of 50% is used.

### 4.3.2 Restaurants and bars

Dine-in data (i.e. meal prices and total days open) was collected from discussions with owners/workers at 11 of the 24 restaurants and bars, but "take-away" (dine-out) revenues are not included. For this category the meal prices do not change with different seasons. The revenue is calculated by multiplying the capacity, average meal price, the table turn, the total days and the occupancy rate. In the equations

below, H, M and L indicate high, middle and low season, respectively.





$$RPD = C * \text{Approx. meal price} * TT \qquad \text{(Eq. 3)}$$

$$RH = RPD * TH * OH \qquad \text{(Eq. 4)}$$

$$RM = RPD * TM * OM \qquad \text{(Eq. 5)}$$

$$RL = RPD * TL * OL \qquad \text{(Eq. 6)}$$

RH, RM and RL are the total revenues for high, middle and low season, respectively. The capacity C, i.e. the total number of people the restaurant can host, is multiplied with the approximate meal price and table turn TT, i.e. number of times a table is occupied with different groups, to calculate the revenue per day RPD.

It is important to notice that, when it comes to the high season H, the number of times a table can be occupied during working hours varies for each restaurant. For example, at Faraglione, which is small but popular restaurant and bar adjacent and open to the port at Levante, one table may turn as much as 20 times during a day because it is open from very early morning until very late at night. However, for middle and low season, the time a table turns is fixed at '1', considering a restaurant never works on full

capacity during these seasons. TH, TM and TL represent the number of total days and OH, OM and OL represent the occupancy rate for high, middle and low season which is 100%, 50% and 15% respectively.

### 4.3.3 Hotels

We were able to collect the required data (capacity, prices, opening season) for 12 out of 17 hotels. As

for the other facilities, there was no official website or they were not open for us to speak with them when data were collected. In the equations below, H and L indicate high and low season, respectively. M1 and M2 represent the two subgroups of middle season.

$$RH = C * PH * TH * OH \qquad \text{(Eq. 7)}$$

$$RM1 = C * PL * TM1 * OM1 \qquad \text{(Eq. 8)}$$

$$RM2 = C * PM * TM2 * OM2 \qquad \text{(Eq. 9)}$$

$$RL = C * PL * TL * OL \qquad \text{(Eq. 10)}$$

RH, RM1, RM2 and RL are the revenues for high season, middle season (M1 and M2) and low season, respectively. In contrast to the seasonal classification of restaurants, hostels and B&B's, the middle season for hotels is divided into two subgroups due to high differences in prices. The first subgroup

includes May and October, while the second one includes June and September. These months are considered together because the price per night per person is more or less the same. When calculating the revenue for the first subgroup, the price by night is the one that is used for the low season even though we are on middle season. TM1 and TM2 are total days for May-October (62) and June-September (60). OM1 and OM2 are the occupancy rate which is estimated as 50% and 60% in

calculations for May and October and June and September, respectively.





### 4.3.4 Leisure activities

While touristic attractions contribute an important amount of revenue to the economy of the island, they close for the low season like B&B's and hostels do, because there are insufficient numbers of tourists to keep the businesses open. The type and price of leisure activities are all determined based on discussions with the owners. The revenue per day (RPD) for seven groups of different activities (i.e. vehicle rentals, scuba diving, snorkelling, kayaking, guided boat tours, boat rental and mudpool) is

calculated for both high H and middle (M1 and M2) seasons.

$$RH = RPDH * TH * OH \qquad \text{(Eq. 11)}$$
$$RM1 = RPD\ M1 * T\ M1 * OM1 \qquad \text{(Eq. 12)}$$
$$RM2 = RPD\ M2 * TM2 * OM2 \qquad \text{(Eq. 13)}$$

TH is the total days of high season (July and August, 62 days), whereas OH is the occupancy rate during

the high season, i.e. the proportions of activities occupied, and RH represent the revenue for high season. The middle season is also divided into two subgroups, as in the case of hotels, but considering different temporal distributions. The first subgroup considers only June (TM1 total days equal to 30) because the revenue is remarkably higher than the total of the rest of the middle season months. The second subgroup consists of April, May, September and October (TM2 total days equal to 122). OM1 and OM2 are the

occupancy rate which is estimated as 90% and 40% in calculations for the first and second subgroups of the middle season, respectively.

## 5 Results

### 5.1 Evacuation effectiveness

**Figures 6a,b** and **7a,b** show simulation results for simultaneous and staged evacuations during low and high seasons, respectively. When considering evacuation time as a proxy for effectiveness, both simultaneous and staged evacuation scenarios are slightly faster during the low season (427 and 401 minutes, respectively) with respect to the evacuation during the high season (535 minutes for both). For the latter one, although both scenarios have equal evacuation times, their evacuation effectiveness differ

(**Figs. 7c**). During the low season, a 95% evacuation effectiveness is reached within 348 and 392 minutes for the simultaneous and staged evacuations, respectively (**Fig. 6c**). For the high season, a similar effectiveness is reached within 365 (simultaneous) and 447 minutes (staged) (**Fig. 7c**). These results have two implications. Firstly, the simultaneous evacuation results in less people left exposed to increasing hazard over time, which confirms findings from previous studies (e.g., Chen and Zhan, 2008;

Jumadi et al., 2019). Secondly, an increase of population of 360% of population between the low and high seasons results only in an increase in evacuation time of ~12%. In fact, assuming that warning time and preparedness time distributions are independent of population size, the main aspects that could impact the evacuation time are the capacity of the boats used for evacuations and the pedestrian speed. For the case of Vulcano, the relatively large capacity of the boats can equally accommodate the increase

of population during the high season and the pedestrian density in the roads considered under both





scenarios does not impact pedestrian speed (the population density in the space, in our case roads, increases beyond 1 person per square meter, which is not reached in Vulcano).

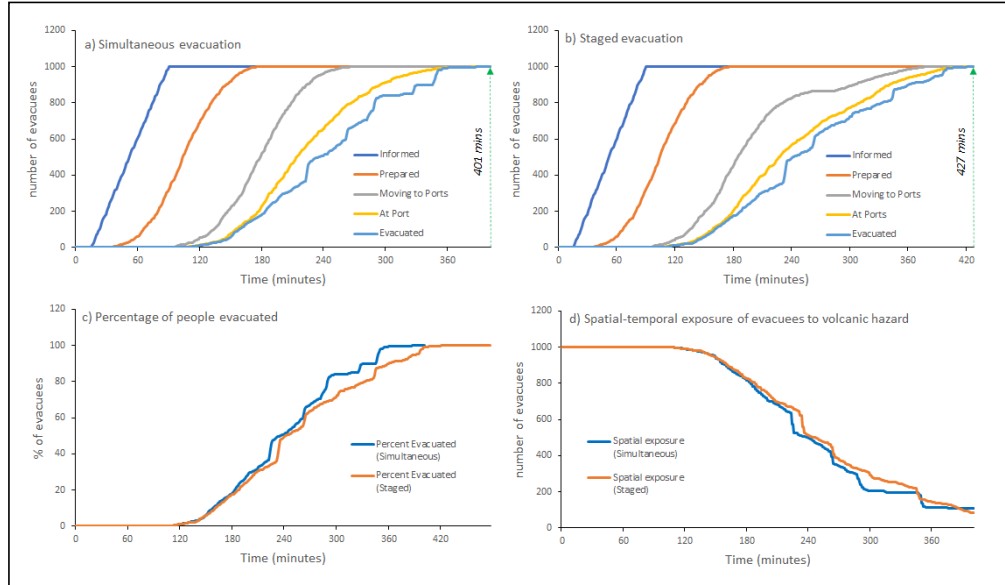

**Figure 6. Plots of evacuation simulations for low-season scenario showing: a) a simultaneous evacuation, b) a staged evacuation, c) percentage of people evacuated with time, d) variation of exposure with time**

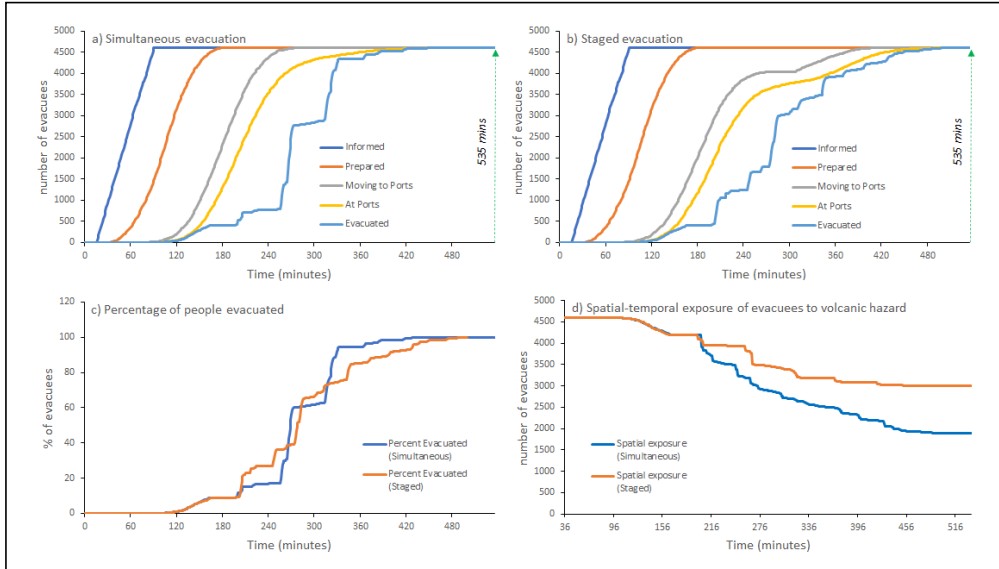

**Figure 7. Plots of evacuation simulations for high-season scenario showing: a) a simultaneous evacuation, b) a staged evacuation, c) percentage of people evacuated with time, d) variation of exposure with time.**




**5.2 Determination of revenue on Vulcano**

With the methodology explained in section 4, the revenues from four different categories (hostels-B&Bs-residences; restaurants; hotels and leisure activities) are calculated. Hotels and restaurants are the only two categories providing revenues during low season (**Table 3**). While calculating the monthly

| Revenues in Low Season (€) | | | |
|---|---|---|---|
| **Business Activity** | **1 Day** | **2 Weeks** | **1 Month** |
| Hotels | 4,943 | 69,195 | 148,275 |
| Restaurants | 1,119 | 15,666 | 33,570 |
| **TOTAL** | **6,062** | **84,861** | **181,845** |

**Table 3 Revenues for each business activity during low season. 14 days are considered for two weeks, while for monthly calculations 30 days are considered.**

revenue by using equations 6 and 10 the number of total days (TL) considered is 30 and the occupancy
rate (OL) is set at 15%. With a monthly amount of 148,275 €, the revenue from hotels is 4.5 times greater than the revenue from the restaurants during low season. From the end of April, the tourist population starts to increase on the island. Equations 2, 3 and 5 are used to calculate the monthly revenues from hostels-B&Bs-residences and restaurants. The number of total days (TM) for a month considered is 30 and the occupancy rate (OM) used for this season is 50%. On the other hand, while calculating monthly
revenue for the middle season for hotels and leisure activities, an average is taken due to different occupancy rates (OM) throughout the season. As seen in equations 8, 9, 12 and 13, the season is divided in two subgroups for these two categories. Thus, first the daily revenues are calculated for each month with designated values, e.g. for leisure activities 90% of occupancy is considered in June whereas 40% of occupancy is considered for September. Then, an average is taken to determine the daily revenue
during middle season. After that, the number of total days (TM) considered to calculate the monthly revenue is 30. As seen in **Table 4**, hotels provide more than half (62%) of the monthly revenue for middle season with a 1,302,927 €, whereas the restaurants, hostels-B&Bs-apartments and leisure activities provide 22%, 10% and 6% of the monthly revenue, respectively.

| Revenues in Middle Season (€) | | | |
|---|---|---|---|
| **Business Activity** | **1 Day** | **2 Weeks** | **1 Month** |
| Hotels | 43,431 | 608,033 | 1,302,927 |
| Hostel, B&B, Residences | 7,112 | 99,561 | 213,345 |
| Restaurants | 14,960 | 209,440 | 448,800 |
| Leisure Activities | 4,067 | 56,936 | 122,005 |
| **TOTAL** | **69,569** | **973,969** | **2,087,077** |

**Table 4 Revenues for each business activity during middle season. Two weeks represent 14 days, while 30 days are considered for a monthly calculation.**

The touristic population reaches its peak point during July and August. Thus, the occupation rate (OH)
is considered 100% for all the categories. The total number of days (TH) for a month is taken as 31, i.e. representing July and August. The revenues are calculated by using equations 1, 3, 4, 7 and 11. Hotels and restaurants provides the highest revenue for this period with 44% and 46% of total revenue




respectively, whereas leisure activities and hostels-B&Bs-residences contribute 4% and 6%, respectively (**Table 5**). While calculating revenue for low and middle seasons, table turn (TT), i.e. the number of times a table is occupied by different groups, is considered 1. However, during July and August, restaurants are full of tourists and a table in a restaurant is served more than once. Thus, the number of TT varies for each restaurant while calculating the revenues for high season.

| | Revenues in High Season (€) | | |
|---|---|---|---|
| **Business Activity** | **1 Day** | **2 Weeks** | **1 Month** |
| Hotels | 174,985 | 2,449,790 | 5,424,535 |
| Hostel, B&B, Residences | 25,357 | 354,998 | 786,067 |
| Restaurants | 184,516 | 2,583,224 | 5,719,996 |
| Leisure Activities | 17,429 | 244,004 | 540,294 |
| **TOTAL** | **402,287** | **5,632,016** | **12,470,892** |

**Table 5 Revenues for each business activity during high season. 14 days are used for two weeks and 31 days, representative for July and August, are considered for a monthly revenue.**

It should be noted that the prices for hotels, hostels-B&B-residences are not constant during different seasons and, in fact, they slightly for each month. The highest prices throughout the year are applied for the second and third week of August which is considered as summer vacation in Italy. An average price for each season is calculated based on website data. Additionally, while calculating the revenues for each season, different occupancy rates are considered to obtain a range of revenues. For example, during low season it has been considered as varying between 5% and 15%. More than half (51%) of the yearly revenue (35,510,782 €) comes from hotels (**Table 6**). The other half is divided between the remaining three groups, with restaurants providing the second highest revenue after hotels with 37% of total revenue.

| **Business Activities** | **Revenues (€)** |
|---|---|
| Hotels | 18,159,925 |
| Restaurants | 13,021,382 |
| Hostels | 2,638,859 |
| Leisure Activities | 1,690,616 |
| **TOTAL** | **35,510,782** |

**Table 6 Total annual revenue for Vulcano Island resulting from hotels, restaurants, B&B's (including hostels and residences) and leisure activities**

### 5.3 Analysis of potential economic impact of an evacuation

**Figure 8a** shows that the total loss of revenue (expressed by the turnover) is significant if the evacuation begins in June and lasts for more than one month (**Table 7**). If it starts in November, the impact becomes significant if it lasts more than 6 months-1 year (i.e. >30 million €). The high season represents the critical period. The impact of an evacuation starting in November and June in the two Vulcano main touristic areas (Porto and Vulcanello) is also considered (**Fig. 8b,c**). A partial evacuation of Piano was not considered because most of the tourist infrastructures are located in Porto and Vulcanello. Clearly the evacuation of only Vulcanello would result in a smaller loss of revenue with respect to a partial evacuation of Porto for any of the durations considered (i.e. <15 milllion euros). However, in the case





of escalating unrest activity, the safety of people is typically prioritized with respect to economic factors.
As a result, the areas that are the most exposed to the hazard (i.e. Porto) would be evacuated first.

| Evacuation | 1 day | 2 weeks | 1 month | 3 months | 6 months | 1 year |
|---|---|---|---|---|---|---|
| *Total evacuation* | | | | | | |
| Starting in November | 6 | 85 | 182 | 545 | 1,945 | 35,525 |
| Starting in June | 99 | 1,393 | 2,885 | 27,827 | 32,023 | 35,525 |
| *Porto only* | | | | | | |
| Starting in June | 58 | 819 | 1,654 | 19,694 | 22,055 | 24,142 |
| *Vulcanello only* | | | | | | |
| Starting in June | 40 | 564 | 1,210 | 7,982 | 9,775 | 11,119 |
| **Table 7. Loss of turnover due to an evacuation (in 1,000 Euro)** | | | | | | |

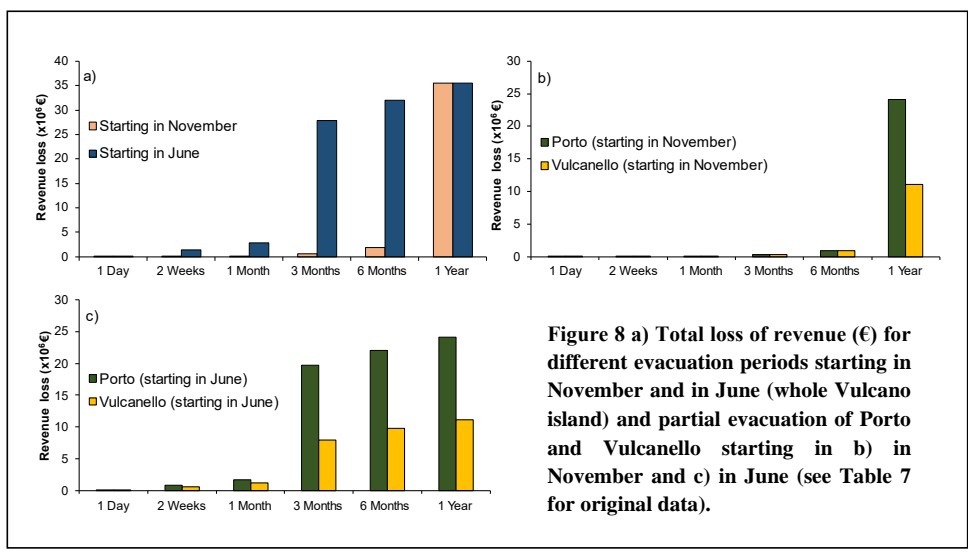

Figure 8 a) Total loss of revenue (€) for different evacuation periods starting in November and in June (whole Vulcano island) and partial evacuation of Porto and Vulcanello starting in b) in November and c) in June (see Table 7 for original data).

## 6 Discussion

### 6.1 Effectiveness of evacuation

The main objective of our study is to provide decisions makers with an operational tool to investigate various evacuation scenarios. This evacuation simulation tool allows emergency managers to identify and optimize individual and organizational parameters (related to actions, behaviours, policies and resources) that minimize the evacuation time as crises evolve. The tool allows to estimate such key indicators as the minimum time necessary to fully accomplish the evacuation which, in the context of volcanic crises, can be compared to eruption forecasts provided by monitoring networks. Together, these two aspects provide a comprehensive picture of the various components to achieve successful emergency management.

However, although the overall evacuation time and the individual evacuation time are vital measures for enhancing the effectiveness of the evacuation process, they do not fully consider the dynamics of hazard and exposure during a volcanic eruption. In volcanic eruptions, hazard and exposure vary in time





and space. In other words, the risk can increase because the probability of eruption might increase with time and because the actual exposure could be significantly higher a few hours after the evacuation order

is issued compared to the first hour due to the movement of people towards the evacuation areas (e.g. ports), which are sometimes closer to the source of the eruption (La Fossa) than where they initiated evacuation (e.g., Vulcanello). Therefore, to reduce exposure the goal should be to evacuate more people faster (Han et al., 2007). The spatial exposure on Vulcano is complicated by to the proximity of the main port (Porto Levante) to La Fossa crater (i.e Porto Levante is located at foot of the northwest flank of La

Fossa). Particularly, for people in Vulcanello moving towards Porto Levante to evacuate requires that they get closer to the hazard source at La Fossa. Evacuating people from these ports can, therefore, increase the exposure in time and space. While optimizing evacuation requires that evacuees move away from the hazard source, evacuation of people in the north side of Vulcano to either Porto Levante or Ponente cannot be done without moving people closer to the hazard source, especially when moving

people to the Porto di Levante because it is closer to La Fossa than Porto di Ponente. Exposure could be reduced by moving people from the Porto di Levante area to Porto di Ponente, but the latter port cannot accept large ships nor handle large volumes of people. It is, in fact, significantly smaller and characterised by shallower water than the port facility at Levante. Therefore, the planning of an effective evacuation should assess the evacuation time as well as the temporal variation of exposure. For the case

of the two evacuation scenarios described above, exposure was assessed based on the distance from La Fossa volcano and was found higher for the staged evacuation (**Fig. 6d**) during both seasons, with an increasing difference over time during the high season (**Fig. 7d**).

Some assumptions have been made to carry out our evacuation simulations that should be mentioned: i) people are not allowed to return to the island after the alarm has been issued, ii) people are only allowed

to evacuate by foot (however, some people might try to drive to ports causing traffic jams and road blocks), iii) people with disabilities are considered in the simulations by using a low walking speed (however, other considerations could be made in order to improve the analysis, iv) the "evacuation preparedness time" includes the time required to organize departure and secure the belongings that are left behind (e.g. house, car(s), other vehicles, boats), v) people might be able to take with them small

pets, vi) animals of farming activities (e.g. goats, cows) are not considered here but represent a critical aspect for an island such as Vulcano, vii) evacuation is carried out from the three ports available on the island (i.e. Porto Ponente, Porto Levante and Porto Gelso) even though the only port that can be accessible by large boats is Porto Levante (more studies should be carried out based on the actual evacuation capacities of Porto Ponente and Porto Gelso, and in various weather and marine conditions).

Finally, while we did not directly include social vulnerability aspects due to small community size and lack of up-to-date data, the current evacuation simulation tool can be enhanced to include social vulnerabilities, especially if it is going to be used in larger and more complex social systems. The simulation can be parameterized based on more granular detail on socio-demographic characteristics of the agent population. This will allow to include social vulnerability factors related to age, health



conditions, gender, language, education, access to resources and information in the evacuation
simulation tool.

**6.2 Assessment of the economic impact of an evacuation of Vulcano island**

The loss of revenue due to touristic business interruption associated with an evacuation of Vulcano

Island is studied as a function of time, in order to investigate the influence of different touristic seasons, and as a function of space, in order to investigate how a partial evacuation affects the economic loss on the island. According to our results, both the time when the evacuation process is carried out and the duration of the evacuation period have significant impact on tourism. For instance, a short-term evacuation (i.e. up to three months) during low season (e.g. November, December) causes less than one

million Euros of revenue loss (about 550,000 €). Should people be evacuated for 6 months, the loss could increase to about 2 million € only after six months due to an overlap with the beginning of the middle season when touristic activities start to resume. One year of total evacuation on the island causes about 35 million € of revenue loss. Only 5% of this loss results from evacuation during low season. This is due to the fact there are no tourists on the island during these months and most touristic activities

ceased. The situation is, therefore, critical if the evacuation needs to be carried out towards the end of the middle season (e.g. June) and/or during the high season when the population on the island reaches its peak point. In such a case, a month-long evacuation in June is almost 1 million € higher than 6 months of evacuation during the low season (i.e. starting from November). After that, a rapid increase in revenue loss is observed on the island: three months of evacuation starting in June causes up to 28 million € of

revenue loss which corresponds to 80% of the one-year loss because it includes the high season.

In addition to the high revenue loss that could occur during the high season, it is important to note that the evacuation process becomes more complicated due to the high number of tourists between June and September (in addition to the diversity of languages represented), whereas an evacuation between November and April concerns only local people, all of whom would presumably speak Italian.

The loss of revenue on the island is also considered as a function of space. To do this, partial evacuations including only Porto or only Vulcanello are evaluated. The main reason for assessing the partial evacuation is to be able to maintain at least some activities on the island, without interrupting all tourism-dependent businesses and also to see which part of the island has the highest impact on the economy. According to our results, during the low season the loss of evacuating Vulcanello is slightly higher than

the loss of evacuating Porto (lower than a million euros). Although most of the touristic facilities and all the restaurants are located in Porto, the largest hotels on the island are all situated in Vulcanello and they are open for the whole year (Therasia Resort Seas and Spa and Jera Residence). However, with the beginning of middle season the revenue loss in Porto exceeds Vulcanello. If the evacuation includes July and August, the loss resulting from evacuating Porto is double of the loss of evacuating Vulcanello.

Piano is not considered in the partial evacuation scenarios. In fact, on this southern side of the island, there are no shops, hotels or any other leisure activities to attract tourists with the exception of a famous



lookout (Capo Grillo) and small beaches. Only two B&B's and two restaurants are located in Piano with revenues negligible compared to those located in Porto and Vulcanello. However, this does not mean that Piano has no effects on Vulcano's economy. As mentioned earlier, the wine factory is situated

between Piano and Gelso. According to the owner, the vineyard flourishes between March and July. Thus, if an eruption occurs during this period and the area is evacuated, there will be at least 60,000 € of loss generating from Piano. Also, the important infrastructures that are not considered in the cost assessment of this study, such as the solar plant located in Piano, may cause problems for other businesses. For example, if the electricity is cut on the island, the restaurants and hotels cannot function

and this affects directly the tourism and thus the revenues, even though the evacuation is partial, and that part of the island is not affected.

Cost assessments are also required to conduct Cost-Benefit Analysis of different mitigation measures. Although evacuating the island will cause an economic loss (i.e. the loss of revenue as the cost), it is a key measure to reduce the impact on public health. It helps ensure the prevention of eruption related

injuries and deaths, hence the main components of benefits. Quantifying the value of life is an ethical issue. Although there are studies that try to assign a value to a human life (e.g. Cropper and Sahin, 2009), here we do not consider it, as this is beyond the scope of our analysis. In any case, if an eruption on the island is imminent, total and/or partial evacuations will be conducted regardless of the cost in order to avoid casualties. However, it is important to evaluate the socio-economic impact on affected

communities for authorities, in order to help them to implement informed decisions.

Although this study has provided some significant findings on the tourist sector of the economic system of Vulcano Island, such as the main income activities and the possible loss in case of an evacuation, it does not provide a complete picture for the cost assessments, and some important caveats need to be discussed. The first and the most important limitation concerns the lack of data. Although the

municipality of Lipari was visited in May 2014, no access was granted for any official data concerning the economic situation of the Aeolian Islands, let alone Vulcano itself. All the data used to calculate the revenue on the island were based on our field visit in May 2014, official websites and various booking websites were used to complete the data set. Another important point to mention is that the main focus of our study is on the revenues originating only from tourism-related businesses, and, therefore, the total

cost of evacuation process is not investigated (i.e. cost of evacuation operations and of relocating people). This loss presents only one part of the total cost associated with an evacuation. An extensive cost assessment requires the consideration of all different types of costs involved with the evacuation process.

**6.3 Potential negative and positive economic consequences of a volcanic crisis**

Forecasting volcanic eruptions and managing volcanic crises presents an important challenge for both scientists (e.g. geochemists, geophysicist, geologists, volcanologists) working in observatories and civil authorities such as those associated with emergency management. In Italy, the Civil Protection


Department play an active role in decision making. Based on the combination of monitoring parameters
provided by the INGV and dedicated Competence Centers and background data available for the event
that might occurre at Vulcano, the Civil Protection Department declares the alert levels in close
collaboration with the Regional Civil Protection authorities. The evaluation is based on the reports of
the phenomena and on the evaluations of hazard made available by Istituto Nazionale di Geofisica e
Vulcanologia (INGV) and by the Consiglio Nazionale delle Ricercheof Italy - Institute for
electromagnetic sensing of the environment (CNR-IREA).

However, many impediments may be encountered in interpreting key aspects such as i) whether or not
unrest will lead to an eruption, ii) the nature of explosive activity (magmatic or hydrothermal), iii) the
eruptive style (i.e., effusive, explosive or both), iv) the potential activation of lateral vents, v) the
eruption magnitude (i.e. erupted volume) and intensity (i.e. the rate of discharge of magma, plume
height), vi) the type, extension and timing of hazards with the potential to impact human life and the
infrastructure supporting evacuation whether occurring either in the unrest phase or eruptive phase or
both. The interpretation of scientific data complicates the decision-making process for the officials
(Fearnley, 2013). Higher levels of scientific uncertainty may thus translate to increased difficulty for
emergency managers to understand the value of evacuation (measured in terms of human lives saved)
and the costs associated with any evacuation that is not accompanied by the occurrence of hazards
necessitating eruption.

When a volcano begins to show increasing signs of unrest above background level, authorities have to
deal with uncertainties and decide how to manage a potential crisis (e.g. have people shelter in place or
evacuate some or all of the population), as scientists cannot guarantee if the unrest will result in an
eruption or not. Although successful forecasts have been made (e.g. Mt St Helen's 1980, USA; Mt
Redoubt 1989-1990, USA; Pinatubo 1991, Philippines), false alarms that cause both scientists and
officials to lose credibility also occurred in the past (Sparks, 2003; Tilling, 2008). For example, during
1983-1985 volcanic crisis at Rabaul Caldera (Papua New Guinea), the government practiced many
evacuation exercises, which led to voluntary evacuations by villagers. They intensified disaster-
preparedness activities when intense earthquake swarms begin to occur in September 1983 and
continued until April 1984. Although there was a high expectation that an eruption was imminent (i.e.
that eruption would take place) by early 1984, the number of earthquake swarms and their intensities
suddenly decreased. The government subsequently dropped the alert level in November 1984 and by
mid-1985 the seismicity returned to its pre-1983 levels (Hastings, 2013). Consequently, the volcanic
crisis resulted in substantial losses of revenue due to business interruptions with the total cost of
emergency preparations exceeding 20 million PNG Kina (~21 million $). At the end, many people
thought that two years of preparation was a waste of money (Hastings, 2013). Nevertheless, some
benefits also emerged from this crisis, as public awareness of potential volcanic hazards increased and
the community became more resilient (Hastings, 2013; Tilling, 2008).



Unfortunately, successful forecasts followed by evacuations may also cause economic distress for communities located in hazardous areas. As an example, in October 1999, almost 19,000 people were evacuated from Baños, Ecuador when Mt. Tungurahua renewed activity after a long period of quiescence. Some 95% of the community's economic activity was dependant on tourism (Lane et al., 2003), showing a similar situation to Vulcano Island. After the evacuation, an economic crisis was felt

both locally and nationally. In the city of Ambato, where evacuees were rehoused, unemployment was an issue, health costs increased by about 103%, and food and beverage prices increased by about 108% (Lane at al., 2003). When authorities realized that economic recovery would be hard without tourists, the tourism industry launched an effective campaign to promote positive views of the area by using the volcano's attractiveness, to convince both domestic and foreign tourists that the situation in Baños was

back to normal. Even journalists were invited to the town to report on the successful recovery (Lane et al., 2003). Finally, in 2000, Baños attracted approximately 23% of the country's 615,000 foreign visitors. In November 2001, 56% of all tourists visiting Baños were foreigners (Lane et al., 2003).
        As seen in both cases at Rabaul and Baños, a volcanic crisis, if not managed well, can easily result in an economic crisis, with or without an evacuation and an eruption occurring. This is also valid in the case

of Vulcano Island. Even without an evacuation order, the increasing level of unrest may cause the local people to leave the island, if they believe that tourism on the island may be affected negatively by the increasing volcanic activity. This is especially true since most business owners are not from Vulcano and they may decide to relocate their activities. In both cases, the economy of the island would be negatively impacted. Additionally, there could be significant negative economic impacts on Vulcano

associated with changes in the volcano alert level even when an eruption or evacuation does not occur, as Peers et al. (2021) described for the protracted unrest at Long Valley Caldera, California, in USA.
        Volcanic unrest and eruptions can also have positive impact on economy. As an example, volcano tourism and geotourism has become more and more popular all around the world. It is estimated that between 150 and 200 million people visit volcanic and geothermal environments on an annual basis

(Heggie, 2009; Erfurt-Cooper, 2011), because a growing number of tourists seek adventure by planning holidays close to active volcanoes (Brace, 2000; Erfurt-Cooper and Cooper, 2010). As an example, in 2008, 1.2 million tourists visited the active volcanic features in Hawaii Volcanoes National Park, 3 million visited the geysers and hot springs of Yellowstone National Park and in 2004 103 million people visited Fuji-Hakone-Izu National Park in Japan (Heggie, 2009; Erfurt-Cooper, 2011). Other than USA

and Japan, geothermal and volcanic activity in Italy and Iceland are also highly attractive destinations for tourists (Heggie, 2009). Research by Bird et al. (2010) in Thorsmork, Iceland near Katla Volcano in 2009 examined the relationship between tourism and volcanic activity. They found that all the participants (tourists) knew that Iceland is volcanically active, but they do not think of volcanic eruptions as hazardous events, hence they lack hazard knowledge. Additionally, most tourists and tourism

employees think that tourism will benefit positively after a future Katla eruption. However, according





to results of Dominey-Howes and Minos-Minopoulos in 2004 in Santorini, Greece, it is the residents who fear that a future eruption may have a negative impact on the tourism.

Vulcano Island appeals to a wide range of tourists: some visit to relax and/or for health reasons, whereas others are attracted to volcanic landform and geothermal features. Thus, an increase of unrest may attract

more adventure-driven tourists, unless such visits are curtailed by civil authorities as a result of increased likelihood of eruption and resulting limitation of the number of people on the island. If the increasing activity on the island results in an evacuation and finally in an eruption, still many tourists interested in natural areas and adventure may want to visit the island once the activity is back to pre-eruption and the risk is decreased. This type of tourism should be foreseen and well organized to boost the local economy

especially after business disruption due to evacuation and/or eruption.

**7. Conclusions**

Evacuation is often the only strategy to save lives in case of extreme volcanic activity and rapidly escalating unrest. This is especially critical for La Fossa volcano whose activity has been characterized

by hydrothermal events, which are typically very sudden and unpredictable, and magmatic events with little warning signals (e.g. 1888-90 Vulcanian cycle). In such a case and considering the high level of population exposure to dangerous hazards, evacuation should be considered even in case of weak unrest. Nonetheless, the timing and routing of evacuation is critical to remove people from the hazardous zone before it is impacted. The Vulcano evacuation simulation tool decribed here has been developed to test

the effectiveness of ABM simulation in evacuation planning for areas subject to volcanic hazards on a small island. Based on a pre-eruption simulation at Vulcano, we have demonstrated that the both the simultaneous and the staged evacuation are slightly faster during the low season (401 and 427 minutes to evacuate 1,000 people, respectively; **Fig. 6a,b**) with respect to high season (535 minutes to evacuate 4,600 people; **Fig. 7a,b**). Nonetheless, we have also shown that the type of evacuation (i.e. staged or

simultaneous) can optimize the number of people evacuated in time, with the simultaneous evacuation being more efficient at removing people from the island than the staged evacuation, especially in the low season (**Figs. 6 and 7**). Additional analyses should be carried out to explore more evacuations conditions (e.g. evacuation by car, evacuation from fewer ports, evacuation after the onset of the eruption) or the role of social vulnerability.

We have also shown how, in an island as Vulcano whose economy is based on tourism, the timing and duration of evacuation can have very different impacts. In fact, if the evacuation of the whole island starts in the low season (e.g., November), the impact becomes significant only if it lasts more than 6 months-1 year, whereas if it starts in June the impact becomes significant after 3 months. In particular, our results show that a total evacuation starting in June, for a period of 6 months or less, result in ~95%

more revenue loss than the evacuation starting in November. This is directly related to the number of high tourist population on the island during that period. In addition, if the evacuation starts in November and lasts for up to 6 months, there is no difference between the partial evacuations of Porto and of





Vulcanello in terms of revenue loss. On the contrary, if the evacuation starts in June instead of November, the revenue loss resulting from evacuating Porto is 30-50% higher than evacuating

Vulcanello. Moreover, for an evacuation lasting more than 6 months (e.g. one year), the result of evacuating Porto causes 50% of higher revenue loss than evacuating Vulcanello. Consequently, we can say that for a partial evacuation, evacuating Porto starting from June will cause the largest impact on the island's economy. This is due to the fact that all leisure activities and restaurants with the majority of hotels, hostels and B&B's are located in Porto. However, it is important to stress that human life has to

be prioritized over economic losses, therefore being the most exposed area to volcanic hazard, Porto should be evacuated even though it is associated with the highest revenue losses. Finally, regardless of the timing of the evacuation and its duration, the total evacuation of the island generates 30-50% more revenue loss than the partial evacuation of Porto, and 45-70% more revenue loss than the partial evacuation of Vulcanello.




**Appendix A**

**Vulcano evacuation simulation tool setting page and interface**

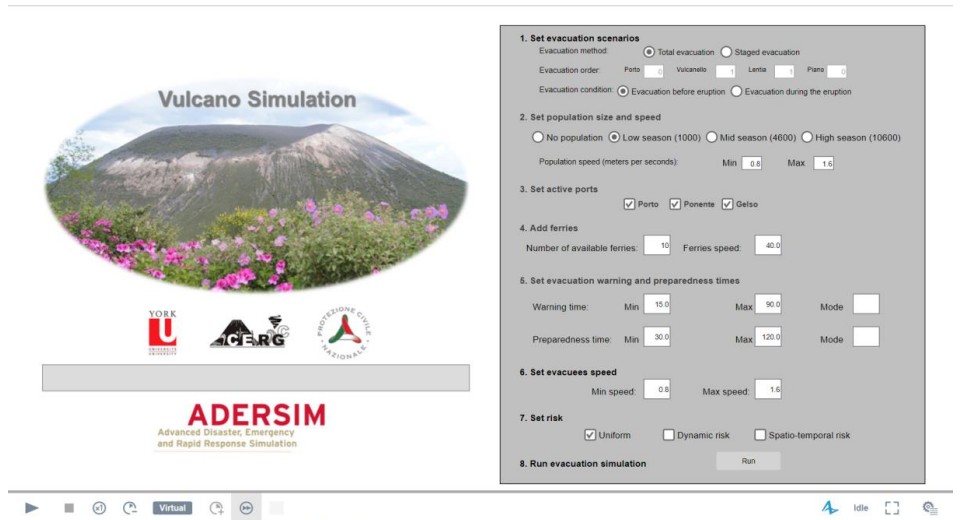

**Figure A1 Parameter and scenario setting page of the Vulcano evacuation simulation tool**


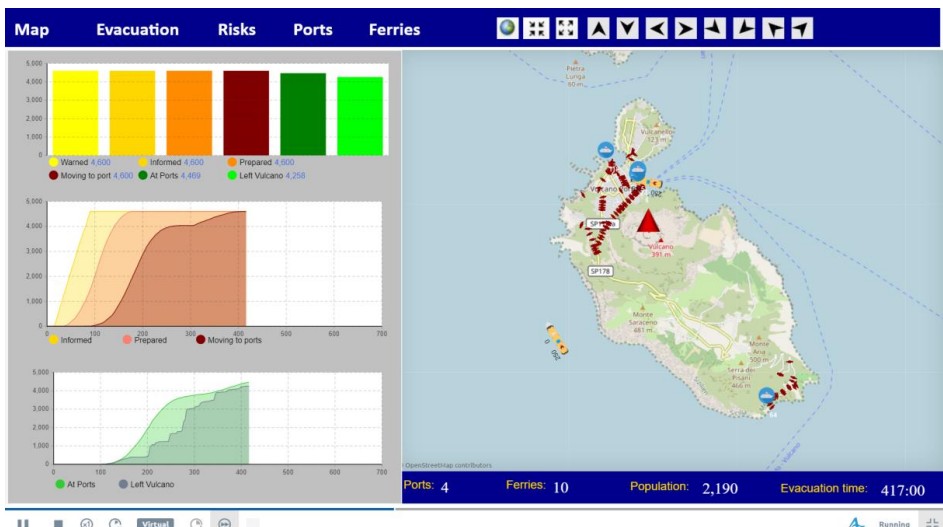

**Figure A2 Main interface page of the Vulcano evacuation simulation tool**



**Appendix B**

**Example of a pedestrian evacuation simulation run (video)**

An example of a pedestrian evacuation simulation run for a simultaneous evacuation scenario during high season on Vulcano (4600 people consisting of 400 people in Piano, 400 people in Vulcanello and 3800 people in Porto; see main text for details) can be found at this link:

http://gofile.me/5ri20/GrlDLHsPQ


The simulation shows movement of evacuees from different parts of the island to their designated or nearest active port from where they will be evacuated by ferry (Porto Levante, Porto Ponente in the North and Porto Gelso in the South). Yellow colour represents people who have received the evacuation order, orange colour shows people who are preparing for evacuation, and brown colour shows people

who are moving towards the ports. Bar at the bottom shows the number of ports and ferries used (in this case 3 ports on Vulcano and 1 port in Milazzo where ferries bring the evacuated people), the number of people at a given time step and the evacuation time indicated in minutes.





**Declarations**

**Availability of data and materials**

The datasets supporting the conclusions of this article are included within the article.

**Authors' contributions**

C.B., A.A., F.R., T.Z. and C.F. have conceived and written the first draft of the manuscript. A.A. has developed the ABM tool and performed the simulations for different evacuation scenarios based on information provided by C.B. and C.F.. T.Z. has carried out the data collection and analysis necessary to the assessment of economic impact of an evacuation of Vulcano under the supervision of F.R., C.B. and C.F.. C.C., M.R., C.E.G., S.B., S.M., M.P. and A.R. have provided key insights

into interpretation of the results. All authors have contributed to the finalisation of the manuscript.

**Competing interests**

The authors declare that they have no competing interests.

**Disclaimers**

The information set out in this publication reflect the author's views.

**Acknowledgements**

The authors are grateful to the colleagues of the Dipartimento della Protezione Civile of Italy and to the participants of the CERG-C program for fruitful discussions that helped shape the evacuation analysis presented in this manuscript. We also thank the population of Vulcano island for their support to the CERG-C program over the years. This work was supported by the Swiss National

Science Foundation (projects #200021-129997 and #IZSEZ0_181030), the CERG-C program of the University of Geneva, and Ontario Research Fund (ADERSIM).




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
