# Peer review of "Assessing the effectiveness and the economic impact of evacuation: the case of Vulcano Island, Italy"

_Natural Hazards and Earth System Sciences, 2021_

## Author Response (AR1)

Dear Editor,

We are pleased to submit a revised version of our manuscript and we thank both reviewers for all their constructive comments. Please find below a detailed reply (red text).

**Reviewer 1**

1) It would be immensely useful to outline clearly the main assumption upon which this work rests. i.e. why the authors believe the Agent-Based Modelling approach used produces results that are sufficiently close to what might be expected in reality to be used as a basis for formulating real-world evacuation plans. Closely related to this would be some discussion of the degree of uncertainty in the modelled outputs of the ABM models. Addressing these two elements directly in the introduction and more discussion of the uncertainty in the results in the discussion section would improve the robustness of this study.

Thank you for your comments and suggestions. In section 2.1 we have described the advantages of ABM:

"Agent-based modelling (ABM) is emerging as a suitable and promising framework for evacuation analysis and planning in recent years (Chen and Zhan, 2008; Liang et al., 2015; Jumadi et al., 2019). ABM is appropriate for modelling complex and interactive systems (Gilbert and Bankes, 2002) such as emergency evacuation because it combines behavioural attributes with spatial and environmental data (Brown and Xie, 2006). Moreover, ABM can provide a more realistic evacuation simulation with respect to aforementioned approaches by incorporating human agents to the geographical environment (Mas et al., 2012; Joo et al., 2013)"

In addition, we have added a few lines in the introduction to explain why we use the ABM and the AnyLogic Software tool and a new paragraph at the end of section 6.1 to discuss the uncertainty issue.

"We developed an Agent-based modelling (ABM) in GIS space using the AnyLogic® software platform to assist emergency managers and assess the effectiveness of specific evacuation parameters, i.e. number of people present on the island (during the low and high seasons), type of evacuation (simultaneous whole community evacuation or sequentially staged evacuation of different areas), eruption probability, exposure, timing (before, during or after the eruptive event). ABM has been used in evacuation simulation extensively (e.g. Bae et al., 2014; Hilljegerdes and Augustijn-Beckers, 2019) and it has many advantages compared to aggregate and static approaches as it allows to incorporate individual level behaviors, event scheduling, dynamics of agent interactions, flexibility, natural description of evacuation process (Mas et al., 2019). In addition, the platform AnyLogic® allows to visually observe and assess the evacuation scenarios."

"Finally, it is important to consider and discuss some stochasticity and uncertainty aspects of the proposed evacuation simulation tool. Given that most of the distributions we have used to describe the various evacuation parameters are uniform, the stochasticity and uncertainty are relatively low, and the different simulations do not produce significantly different results. The

main source of uncertainty in our model is related to the random distribution of population and capacity of the ferries. However, more parameters can be varied in order to explore a wider range of conditions.."

2) Section 4.3 Should mention that using turnover overestimates the economic loss. The economic loss would be the amount of profit lost for the owners. However, for the employees, the loss would be equivalent to lost wages. Turnover is however more transparent, easily calculable and is a reasonable proxy for economic loss.

Thanks for this comment. In fact, there could be confusion with this terminology. We have added a sentence in section 4.3 "Methodology to calculate the revenues from touristic business activities in Vulcano" to highlight the difference between the concept of "revenue" used in our paper (in fact, the turnover) and the concept of "added value" used by the national account* (which is a more relevant concept, but the data to estimate it is not available in most of the small islands). Contrary to the turnover, the added value does not include the value of the intermediate goods and services (for example the food and energy that a restaurant must buy), but includes the profits, wages, interests and amortizations.

* https://unstats.un.org/unsd/nationalaccount/data.asp

Added text: This revenue must not be confused with the added value provided by the national accounts, which includes the profits, wages, interest and amortizations, but not the intermediate goods and services.

3) Explicit mention of the volcanic hazards posed by Vulcano and a hazard map for the island would provide useful context.

Thanks for this comment. We have added a clarification of the most likely hazards associated with La Fossa volcano in the section 3.1 Geological settings and implications for evacuation planning. We have also added some description of existing hazard maps for tephra fallout, ballistics, PDCs and lahars at the end of section 4.1.1. However, we prefer not to include a hazard map for the island as the evacuation is assumed to be issued before the beginning of the eruption.

4) In Conclusions: clarify how these projections should be used. Not to delay evacuations but to model varying impacts for different scenarios to enable proper allocation of resources required for evacuations and economic support of the affected areas.

Very good point. This was added to the conclusion section. Thank you.

Comments by line number:

 25 be quantitative rather than just stating "more efficient"

More details have been added to the abstract:

"As an example, even though the overall duration is similar for both evacuation strategies, after 370 minutes about 96% of people would be evacuated with a simultaneous evacuation, while only 86% would be evacuated with a staged evacuation during the high season."

30 be quantitative rather than just stating "significant economic impact"

We added more details to the abstract:

"We also present a model to assess the economic impact of evacuation as a function of evacuation duration and starting period that reveals that in case an evacuation that lasted 3 to 6 months was initiated at the beginning or at the end of the visitor season, it would cause a very different economic impact to the tourism industry (about 2-8% and 74-86% of the total annual turnover, respectively)."

51 other natural disasters can also have significant and useful warning times. Large storms for example.

We replaced this sentence with:

"In this regard, volcanic crises differ from many other natural hazards as they are often associated with an unrest phase, during which most volcanic systems exhibit precursors from hours to days, weeks, and even months before the onset of an eruption"

284 "divided"

Corrected. Thanks

302 Ho, 1992 doesn't seem to be in the reference list.

Added to the reference list

353 what happens in the case of a chaotic panicked evacuation?

This is a good question. Chaotic panicked evacuation could happen especially if the evacuation begins during an eruption. In our simulation we assume that the evacuation is taking place before the eruption starts, and, therefore, we assume that is not chaotic.

363 Are there any hazards that could impact the ferry in port? I.e. should people be considered evacuated only once the ferry leaves the port?

Good point. In the results presented here we count evacuees when the board the ferry, but we also provide the numbers for evacuees who left the port. For ports close to the La Fossa, this should be a consideration, especially if evacuation occurs during the eruption. We added a note at the end of section 4.1.2 to clarify this:

"Given that our simulations are based on the assumption that the evacuation takes place before the eruption, once the ferries arrive, evacuees board and they are considered to be evacuated. However, in case the evacuation was carried out during the eruption, people should be considered evacuated once the ferries actually leave the ports, as both ports and ferries could be impacted by the eruption."

373 can any volcanic hazard interfere with ship speed. tephra fall etc. Can volcanic hazards disrupt communication?

Volcanic hazards would certainly disrupt evacuation. In our paper Bonadonna et al. (2021) (https://doi.org/10.1186/s13617-021-00108-5) we address these aspects. As an example, possible communication signal attenuation (e.g. radio) is expected for tephra accumulation larger than 1 cm (Jenkins et al., 2014, 2015). Potential damage to maritime infrastructure is better assessed based on sedimentation rate more than tephra accumulation, with 1 g/m2/h causing speed reductions and increased restrictions on vessel numbers in harbours and 500 g/m2/h causing most vessels to stop functioning due to impaired visibility (Blake et al. 2017a). However, in these simulations we consider evacuation prior the onset of eruption. Simulations of evacuation during an eruption would require different assumptions and conditions.

 Table 2 Are there any hydrofoils available? What is the hydrofoil speed? They are mentioned under capacity and then not referred to again

Sorry for the confusion. In fact, hydrofoils were considered as the low end of ferry capacity (200 passengers). However, for sake of simplicity, we removed reference to hydrofoils and only left ferries with different capacity (200, 400, 600, 800)

Figure 5 Porto Levante, Porto Ponente in the North and Porto Gelso all need to be labelled clearly on the map.

We replaced the figure 5 to include this information

442 delete "on the contrary"

 Deleted

590 Do you have some hypothesis about why this is the case, why does a 360% increase in population only lead to a 12% increase in evacuation time? Add to the discussion.

This is mostly due to the fact that the number of people to be evacuated both in low and high season can be well managed by the capacity of ferries used (200 to 800 passengers). We have added some additional explanation in section 5.1:

"Secondly, an increase of population of 360% of population between the low and high seasons results only in an increase in evacuation time of ~12%. In fact, assuming that warning time and preparedness time distributions are independent of population size, the main aspects that could impact the evacuation time are the capacity of the ferries used for evacuations and the pedestrian speed. For the case of Vulcano, the relatively large capacity of the boats can equally accommodate the increase of population during the

high season and the pedestrian density in the roads considered under both scenarios does not impact pedestrian speed (the population density in the space, in our case roads, increases beyond 1 person per square meter, which is not reached in Vulcano). If a larger number of people had to be evacuated (e.g., 10,000 people as supposed to 4,600), the time needed to evacuate 95% of the population would nearly double because of the number of ferries (10) and associated capacity (200 to 800 passengers) set in the simulation. However, 1,000 people (considered in the low-season scenario) and 4,600 people (considered in the high-season scenario) can be almost equally managed by the capacity and number of ferries used. "

Table 3 include location "Vulcano" in the table or caption

Added

There appear to be small arithmetical errors (rounding errors?) in table 3 e.g. 4943*30 = 148290, not 148275   please  recheck the arithmetic and correct it.

All numbers in Tables 3, 4, 5, 6 and 7 have been revised and corrected. Thanks for noticing these small discrepancies.

Table 4

include location "Vulcano" in table or caption

Added

There appear to be small arithmetical errors (rounding errors?) in table 4 please recheck the arithmetic and correct it.

All numbers in Tables 3, 4, 5, 6 and 7 have been revised and corrected. Thanks for noticing these small discrepancies.

Table 5

include location "Vulcano" in table or caption

Added

There appear to be small arithmetical errors (rounding errors?) in table 5 please recheck the arithmetic and correct it.

All numbers in Tables 3, 4, 5, 6 and 7 have been revised and corrected. Thanks for noticing these small discrepancies.

Table 7 recheck arithmetic

All numbers in Tables 3, 4, 5, 6 and 7 have been revised and corrected. Thanks for noticing these small discrepancies.

796 should 'eruption' be 'evacuation'?

rephrased

871 delete extra 'the' "…we have demonstrated that the both the simultaneous…"

rephrased

875 '…simultaneous evacuation being more efficient at removing people from the island than the staged evacuation, especially in the low season…' include numerical values which demonstrate increased efficiency.

We added some values:

"In fact, after 300 minutes about 84% and 72% of people would be evacuated with a simultaneous and a staged evacuation, respectively, during low season, and after 370 minutes about 96% and 86% would be evacuated with a simultaneous and a staged evacuation, respectively, during the high season (Figs. 6c and 7c)."

880 remove as "…We have also shown how, in an island **like** Vulcano, whose economy is based on tourism,…"

replaced

881 – 883 include numerical values which illustrate the differential impacts being described.

We added some values:

"We have also shown how, in an island as Vulcano whose economy is based on tourism, the timing and duration of evacuation can have very different impacts. In fact, if the evacuation of the whole island starts in the low season (e.g., November), the impact becomes significant only if it lasts more than 6 months (> 8% of annual total turnover), whereas if it starts in June the impact becomes significant after 1 month (> 5% of annual total turnover) and reaches 74% of the annual total turnover after 3 months."

884 "results" not result

Not sure we understand the issue here

885 an evacuation rather than "the evacuation".

 changed

885 delete "number of"

rephrased

886 for improved clarity state the period explicitly again rather than  just "that period"

rephrased

891 delete "of"

 deleted

**Reviewer 2**

Line 242, line 317: Gas emissions should be also mentioned (see the recent crisis of Vulcano in September-October 2021).

Gas emissions have been highlighted in section 3.1. Geological settings and implications for evacuation planning

Line 649 and 730-739: Regarding partial evacuation it should be considered that, according to what is reported in lines 194-199, the area of Port hosts some critical infrastructures as the main power plant, ie: not only "touristic infrastructures".

Of course, most critical infrastructures are located in Porto. However, here we mention the touristic infrastructure because the economic assessment is based on economic turnover related to tourism

Line 689: The evacuation by foot does not allow transporting heavy baggage or other goods (such as the car). This could be accepted by the people in case of imminent risk for life, but probably not in case of a preventive evacuation. Are you considering the evacuation by foot for simulation purposes (ie: a model approximation), or you consider that this is, in any case, the best solution? Since this could be a critical point for the evacuation model, probably it needs a better description.

Thank you for the very good comment. We assume that evacuation by foot is the main evacuation mode for the sake of simulation. However, a combination of evacuation strategies (both by foot and motorized vehicles) could also be and will certainly be considered in the future with this simulation tool.

Line 691: "people with disabilities are considered in the simulations by using a low walking speed". This seems not very realistic, considering that elderly or disabled people could have serious walking problems. This approximation can be considered for simulation purposes but probably it is not realistic for an evacuation plan.

We agree with this comment. For the sake of these preliminary simulations, we decided to make some easy assumptions such as this one. However, we agree that for a more accurate simulation, more specific assumptions for people with disabilities should be made (e.g. integrating evacuation with motorized vehicles). We have clarified this point in section 6.1:

iv) people with disabilities are considered in the simulations by using a low walking speed; however, other considerations could be made in order to improve the analysis (e.g., integrating evacuation with dedicated motorized vehicles),

Line 711: I think that a partial evacuation of a given area could have also a psychological impact on the tourists and residents living in neigbouring areas, triggering, probably, a spontaneous evacuation of other zones. Did you consider this situation?

This can also be a possibility not considered in the current simulations. We have added a clarification in section 6.1:

viii) people follow the instructions provided in the evacuation orders (this is particularly important for staged evacuation as people in each community are asked to evacuate according to their turn; the possibility of having a fraction of the population not following the order of staged evacuation can be included in the simulations in order to add a level of uncertainty)

Line 715: Please, specify here if you are considering only the costs related the activities indicated in Tables 4-6 (which do not consider other possible turnover related to shops, transports, services, etc.)

We have specified in the caption of Table 7 that these data are based on Tables 3, 4 and 5

Lines 713-720: It should be considered that an evacuation during the "low season" could affect or compromise also the "high season", due to the typical maintenance works of the touristic infrastructures performed during the low season and the impact on the activity of hotel booking, etc.

This is an interesting remark. We added a paragraph to address this aspect:

It should also be considered that an evacuation during the low season could affect or compromise also the high season, due to the typical maintenance works of the touristic infrastructures performed during the low season and the impact on preparation touristic activity (e.g. hotel booking). However, eruptions also attract tourists, as recently shown by the 2021 crises of Cumbre Veja (La Palma, Spain) and Fagradalsfjall (Iceland). As a result, the overall impact on the high season revenue of an evacuation during the low season due to an eruption of La Fossa would be difficult to forecast.

Lines 873-874. Considering the uncertainties, expressing the evacuation times in minutes (with 1 minute resolution) seems not very realistic. Probably reporting times in hours with almost one decimal digit could sound better.

Thanks for this comment. We now provide the time both in minutes and hours.

Typos:

Lines 250 and 784: Ricercheof -> "Ricerche of" (insert space)

corrected

The citation "Bonadonna et al., 2021" is ambiguous since the References report two papers with this reference. Probably you could resolve the ambiguity by labelling them 2021a and 2021b.

We removed one of the references that is not relevant for this paper